# Evaluation of Environmental Radioactivity and Hazard Impacts Saudi Arabia Granitic Rocks Used as Building Materials

Othman Fallatah [1],* and Mahmoud R. Khattab [2]

1    Department of Nuclear Engineering, Faculty of Engineering, King Abdulaziz University, P.O. Box 80204, Jeddah 21589, Saudi Arabia
2    Geochemical Exploration Department, Nuclear Materials Authority, El-Maadi, Cairo P.O. Box 530, Egypt
*    Correspondence: ofallatah@kau.edu.sa

**Abstract:** People use granite in residential buildings on a large scale all around the world. Knowing granite's radiological characteristics allows for the safe use of its properties to be maintained over time. Our findings are significant for two reasons: first, they may increase public awareness of the naturally radioactive properties of the materials under investigation, and second, they are necessary to establish standards, regulations, and management for the building materials used in Saudi Arabia. In this study, twenty-four granitic rock samples were collected from the Hai'l area in Saudi Arabia, and these samples are used as building material. Gamma spectrometry was used to measure the amounts of $^{238}U$, $^{232}Th$, $^{226}Ra$, and $^{40}K$ in the collected granitic rock samples. The obtained data was also used to calculate some environmental hazard parameters, such as the absorbed gamma dose rate (AGDR), annual effective dose rate (AEDR), radium equivalent ($Ra_{eq}$), external and internal hazard indexes ($H_{ex}$ and $H_{in}$), gamma index ($I_\gamma$), alpha index ($I_\alpha$), and excess lifetime cancer index (ELCR). We concluded that the collected granitic samples are harmful and need more attenuation if used as building materials.

**Keywords:** building materials; radionuclides; granites; rocks; Saudi Arabia





## 1. Introduction

Natural radioactive materials are components that exist naturally and are radioactive. Examples of these substances include rocks, soil, water, and air. Natural radionuclides are a result of how the Earth was formed; hence, there is no way to get rid of them. Calculating the effects of radiation exposure from terrestrial and interplanetary sources requires an understanding of radioactivity distribution and ambient radiation levels. Despite the fact that these radionuclides are widely spread, their concentrations are influenced by local geological conditions, which differ from one location to another [1].

Exposure to ionizing radiation for humans is unavoidable. The primary exposure sources are cosmogenic and terrestrial radioisotopes. The isotopes $^{238}U$, $^{232}Th$, and $^{40}K$ are found in all components of the environment, including air, water, food, soil, rock, and building materials [2]. Around 85% of the radiation dose that the overall world's population is exposed to is caused by these radioisotopes, which are found in construction materials. As a result, it's important to keep an eye out for contamination brought on by their radioactivity in homes and to share information on the concentrations and distribution of terrestrial radioisotopes in building materials [2,3]. The soil and rocks of the earth are a common source of three different types of construction materials: structural materials, covering materials, and additive raw materials. Structural materials such as cement, concrete, mortar, clay bricks, etc. are frequently used for building structures. The additive raw materials, such as fly ash, bauxite, and phosphogypsum, are used as optional components to change some of the properties of building materials, whereas the covering materials, such as granite, ceramic, and marble, are used for decoration and insulation [4]. Determining the

natural radioactivity levels of building materials is essential for assessing the radiological risks associated with radiation exposures as well as for defining national standards and guidelines for these materials according to international recommendations. There has been a significant increase in interest in researching the natural radioactivity of construction materials and their effects on the general public recently due to growing social concern [5].

The radiation exposure from natural radionuclides in building materials is influenced by a number of factors, including the location and style of the residences as well as ventilation habits. Natural radionuclide activity concentrations in building materials are also influenced by the geological and geochemical characteristics of the study area [6].

The concentrations of radionuclides in the soil of each part of the Earth's crust are principally responsible for the natural radioactivity level and its external exposure outdoors as a result of terrestrial gamma-ray emissions. In general, igneous rocks such as granitic rocks have higher levels of radioactivity than sedimentary rocks due to their higher content of radioisotopes of thorium, uranium, and potassium, with the exception of phosphate rocks and some shales, which have relatively high concentrations of these radioisotopes [7].

Certain quantities of terrestrial radioisotopes are typically present in geological materials, such as granitic rocks utilized in industry and the industrial products developed from them, with varying concentrations depending on the origin of the rocks. Therefore, it is critical to have a thorough understanding of the concentrations and distributions of terrestrial isotopes in rocks. This will enable us to prevent using geological materials with a high content of terrestrial radioisotopes that could result in environmental contamination from natural radioactivity [8].

The main aim of this study is to survey and measure the natural radioactivity due to the presence of $^{238}$U, $^{232}$Th, $^{226}$Ra, and $^{40}$K in commonly used granitic rocks that are used as building materials and determine environmental hazards and their impacts in the Hai'l area of Saudi Arabia. A radiological analysis for the Hai'l area is suggested by the current study, which also maps background radiation levels. This map will be used to look for any variations in background radiation levels brought on by geological processes or other radiation-related variables. The objective of the current research was also to evaluate the gamma radiation exposure caused by granitic rocks. Estimating a few of the radiological hazards' criteria allows one to determine the level of radioactive threat.

## 2. Materials and Methods

### 2.1. Geological Settings

Twenty-four granitic rock samples were collected for this study. The samples were selected depending on some factors, such as accessibility and utilization. The Ha'il area, located between latitudes 27°00′–28°05′ N and longitudes 41°00′–42°15′ E, encompasses around 16,500 km² of the Arabian Peninsula's northern section (Figure 1). Conditions are arid and desert-like. The majority of the yearly rainfall, which amounts to a few tens of centimeters and occurs between November and March, forms brief lakes that dissipate and leave behind tiny sabkhahs. Wintertime temperatures rarely get above 10 °C during the day and are typically below 0 °C at night. Northerly winds are the primary direction. The season, which lasts from April to October, is characterized by southerly and southwesterly winds and commonly experiences days with highs over 40 °C. The enormous An-Nafud and the Arabian Shield, which are portions of the Ha'il region and connected to it by the Rub' al Khali in southern Saudi Arabia, are notable for their varied topography and geomorphology. A substantial chunk of the Ha'il region is made up of the huge An-Nafud, a sand-filled depression covering over 64,000 km². Its distinctive characteristics include the limestone sand and steep wadis of the Arabian Shield, which ascend to hills. However, sandstones from the Paleozoic and Mesozoic appear to have secondary or often tertiary origins for the sand seen in sand dunes and sand sheets [9].

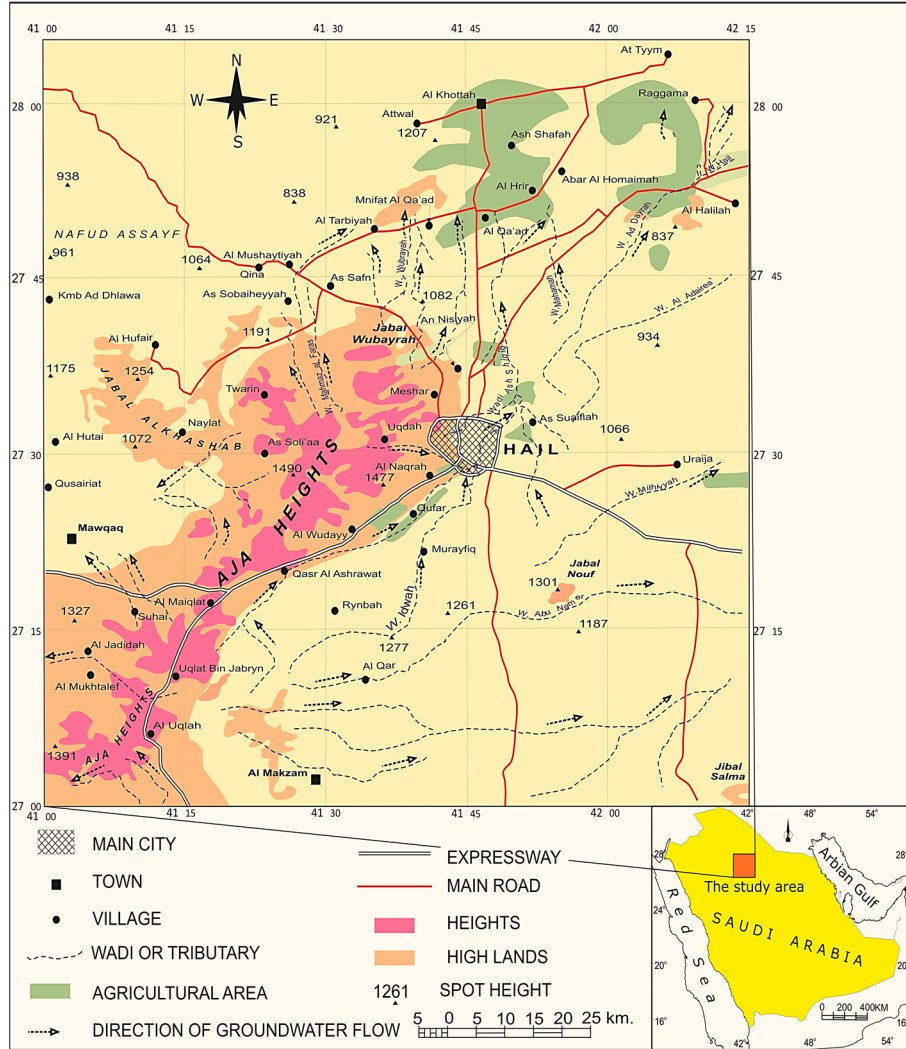

**Figure 1.** Sample locations and geological map of the studied area after Hereher et al. [10].

Quaternary surficial deposits in the high radiation area cover Hail quadrangle's Phanerozoic bedrock and some of the Proterozoic basement. Eolian sand, with minor amounts of gravel, alluvium, and sabkhah, is the main component of the deposits. The Cambrian to Early Silurian succession of mostly sedimentary rocks and late Proterozoic volcano sedimentary and intrusive rocks with complicated geology underlie the study area. The Proterozoic rocks are found in the area's southwest and mostly consist of relatively recent granitic intrusions, such as monzogranite and the more advanced alkali-feldspar granites of the Abanat suite, which are primarily found as large batholiths in the Hail quadrangle's center, where they form the topographically prominent Aja massif (Figure 1).

### 2.2. Analytical Methods

The collected granitic rock samples were analyzed using γ-spectrometry with a relative efficiency of about 60%. By utilizing efficiency-specific radionuclide techniques and the gamma photon's self-absorption effects, the uncertainty of gamma-ray intensities was decreased [11]. From γ-lines of its daughter, $^{234}$Th and $^{234m}$Pa, respectively, are used for the measurement of activity concentration of $^{238}$U at 63.3 and 1001 KeV [12–14].

Utilizing the γ-lines of $^{228}$Ac (338.4 keV and 911.2 keV) and the γ-lines of $^{208}$Tl (583 keV and 2614.4 keV), the specific activity concentration of $^{232}$Th was determined [15]. The average activity of its daughters, $^{220}$Rn, $^{214}$Pb, and $^{214}$Bi, is used as a proxy for the activity concentration of the $^{226}$Ra determination [16]. The activity concentration of $^{40}$K was directly measured via its -line at 1460.8 keV.

### 2.3. Environmental Hazard Impacts

Since the absorbed gamma dose rate is intimately related to the radiological and clinical effects, calculating it is an essential first step in determining the possible risk to one's health. The dosages of $^{226}$Ra, $^{232}$Th, and $^{40}$K are created from the measured activity concentrations using conversion factors of 0.461, 0.603, and 0.0414, respectively [17]. The equation below can be used to calculate the absorbed gamma dose rate, AGDR, in nGy h$^{-1}$ using these variables:

$$AGDR = 0.461\ CRa + 0.623\ CTh + 0.414\ CK \tag{1}$$

where $C_{Ra}$, $C_{Th}$, and $C_K$ are the activity concentrations; Bq.kg$^{-1}$ of $^{226}$Ra, $^{232}$Th, and $^{40}$K.

To calculate the annual effective dose equivalent, the observed value of the airborne dose rate is employed. The conversion coefficient between the rate of airborne dose absorption and the effective dosage equivalent received by an adult must be taken into consideration in order to support these findings [17]. The values of these two variables are influenced by the region's climate and the population's average age. The annual effective dose rates are calculated using the following conversion ratio: 0.7 Sv Gy$^{-1}$ between the absorbed dose in the air and the effective dose, as well as the outdoor occupancy factor of 0.2 [17,18]. The following equation is used to calculate the annual effective dose rate [19]:

$$AEDR\ (mSv.yr^{-1}) = AGDR\ (nGy.h^{-1}) \times 8760\ h\ yr^{-1} \times 0.7 \times (10^3 mSv.10^{-9})\ nGy \times 0.2 = D\ (mSv.yr^{-1}) \times 1.23 \times 10^{-3} \tag{2}$$

Radium equivalent activity, which was suggested as a single quantity to reflect the various activities of $^{232}$Th, $^{226}$Ra, and $^{40}$K, was proposed in order to study the radiation dangers associated with it. It is predicted that employing 1 Bq.kg$^{-1}$ of $^{226}$Ra, 0.7 Bq.kg$^{-1}$ of $^{232}$Th, or 13 Bq.kg$^{-1}$ of $^{40}$K will produce gamma rays at the same dose rate. The highest safe dose value for Ra$_{eq}$ is 50 Bq.kg$^{-1}$ [19].

The radium equivalent (Ra$_{eq}$) was determined using the following equation:

$$H_{in} = C_{Ra} + 1.43 \times C_{Th} + 0.077 \times C_K \tag{3}$$

The external hazard index (H$_{ex}$) of the released gamma radiation is used to evaluate the external risk. This equation is used to calculate it [20]:

$$H_{ex} = C_{Ra}/370 + C_{Th}/259 + C_K/4810 \tag{4}$$

where $H_{ex}$ is the external hazard index and, $C_{Th}$, $C_{Ra}$, and $C_K$ are the activities of $^{232}$Th, $^{226}$Ra, and $^{40}$K in Bg.kg$^{-1}$, respectively.

Internal exposure to $^{222}$Rn and its radioactive progeny is controlled using the internal hazard index (H$_{in}$). It can be calculated using the equation below [20]:

$$H_{in} = C_{Ra}/185 + C_{Th}/259 + C_K/4810 \tag{5}$$

where $H_{in}$ is the external hazard index and, $C_{Th}$, $C_{Ra}$, and $C_K$ are the activities of $^{232}$Th, $^{226}$Ra and $^{40}$K in Bg.kg$^{-1}$.

The γ-radiation level index is performed as a scanning tool to identify materials that could become of health interest when used as building materials. The value of the γ-index must be less than unity to keep the radiation risk to a minimum, and the radiation exposure from construction materials due to radioactivity must be limited to 1.5 mSv$^{-1}$. By using the following equation, the γ-index is calculated [21];

$$I_{\gamma} = C_{Ra}/150 + C_{Th}/100 + C_K/1500 \tag{6}$$

where $I_{\gamma}$ is γ-radiation level index and $C_{Th}$, $C_{Ra}$, and $C_K$ are the activities of $^{232}$Th, $^{226}$Ra, and $^{40}$K in Bg.kg$^{-1}$.

Excess lifetime cancer risk, or ELCR, a radiological indicator, is used to estimate the probability that a person might get cancer as a result of prolonged drug exposure. Results from the yearly effective dose at a particular exposure level enable us to estimate the risk of cancer. It is defined as the number that, given a certain number of people exposed to a carcinogen at a specific dose, can be projected to occur as new malignancies:

$$ELCR = AED \times DL \times RF \tag{7}$$

where AED is the annual equivalent dose, DL is the average duration of life, which is evaluated to be 70 years, and RF is the risk factor ($Sv^{-1}$), i.e., the fatal cancer risk per Sievert.

The alpha index; $I_\alpha$ or internal health index, which is related to the assessment of excess $\alpha$-radiation due to the inhalation of $^{222}$Rn escaping from building materials, was calculated from the following equation [22]:

$$I_\alpha = A_{Ra}/200 \tag{8}$$

## 3. Results and Discussion

### 3.1. Radionuclides Activity Concentrations

The activity concentrations of $^{238}$U, $^{232}$Th, $^{226}$Ra, and $^{40}$K were measured in the collected granite rock samples and listed in Table 1 in $Bq.kg^{-1}$. $^{238}$U activity concentrations were ranged between 119 and 532 $Bq.kg^{-1}$ with an average of 318.6 $Bq.kg^{-1}$, $^{232}$Th was varied between 47 and 1058 $Bq.kg^{-1}$ and averaging 486.8 $Bq.kg^{-1}$. However, the activity concentrations of $^{226}$Ra and $^{40}$K ranged between 19 and 255 $Bq.kg^{-1}$ with an average of 102.5 $Bq.kg^{-1}$ and between 135 and 1519 $Bq.kg^{-1}$ with an average of 726 $Bq.kg^{-1}$, respectively (Table 1). Furthermore, the average values of radioisotope ($^{238}$U, $^{226}$Ra, $^{232}$Th, and $^{40}$K) concentrations obtained were above the worldwide average of these radionuclides in building materials, which is 50, 50, 50, and 500 $Bq.kg^{-1}$ for $^{238}$U, $^{232}$Th, $^{226}$Ra, and $^{40}$K, respectively [6]. The correlations between $^{232}$Th activity concentrations and $^{238}$U activity concentrations derived from the slope in Figure 2 were 0.0323, which is lower than the global ratio of 0.3 [19]. The potassium-thorium cross plot is widely used to recognize clay mineral associations and to discriminate micas and feldspars. Furthermore, in Figure 2, we noticed a strong correlation between $^{40}$K and $^{232}$Th activity concentrations with r = 0.912; the best-fitting relation is again of a linear type. Thorium (through adsorption) and potassium (through chemical composition) are both associated with clay minerals, making the ratio eTh/K a diagnostic marker of other radioactive minerals. Table 1 presents the results for the Ha'il area, in which the eTh/K ratio ranged from 20.4 to 94. These values are much higher than $2 \times 10^{-4}$, which indicates that this area's granite is fresh. There is a low correlation between $^{40}$K and $^{238}$U results, equaling 0.265. In Figure 2, we can see that uranium is present as fixed uranium due to the presence of refractory minerals.

The U ratio in the studied rock samples was of an order higher than 3.5 [23]. If this ratio is less than 3.5, that means uranium enrichment, whereas uranium migrates out (depletion of U), which is indicated by a higher ratio of more than 3.5. For the studied samples, the Th/U ratio varied between 0.7 and 100.1, with an average of 4.5; 50% of the collected samples have a Th/U ratio lower than 3.5, indicating an accumulation of uranium (migration in). On the other hand, 50% of the collected samples have a Th/U ratio higher than 3.5, indicating enrichment of uranium (migration out) (Figure 3).

Enrichment of such authigenic U can be calculated by the following equation [24]:

$$(Ua) = (U - Th)/3 \tag{9}$$

where the square brackets represent the concentration expressed in ppm. Most values of authigenic uranium are negative, indicating uranium migration out, and the positive values indicate the presence of authigenic U (Figure 3).

**Table 1.** Radionuclides activity concentrations $^{238}U$, $^{232}Th$, $^{226}Ra$ and $^{40}K$ (Bq.kg$^{-1}$ and ppm) as well as activity ratio in the collected granitic rock samples.

| Samples | $^{238}U$ Bq.kg$^{-1}$ | $^{232}Th$ Bq.kg$^{-1}$ | $^{226}Ra$ Bq.kg$^{-1}$ | $^{40}K$ Bq.kg$^{-1}$ | $^{238}U/^{226}Ra$ | $^{232}Th/^{238}U$ | $^{238}U$ ppm | $^{232}Th$ ppm | $^{226}Ra$ ppm | $^{40}K$ ppm | Authogenic U (U-Th/3) | eU/eTh | eTh/eU | eU/eRa | eTh/K |
|---|---|---|---|---|---|---|---|---|---|---|---|---|---|---|---|
| 1 | 348 | 671 | 148 | 964 | 2.4 | 1.9 | 28.1 | 166.1 | 13.3 | 3.1 | −46.0 | 0.17 | 5.92 | 2.1 | 53.6 |
| 2 | 452 | 680 | 152 | 980 | 3.0 | 1.5 | 36.5 | 168.3 | 13.7 | 3.1 | −44.0 | 0.22 | 4.62 | 2.7 | 54.3 |
| 3 | 453 | 775 | 153 | 1110 | 3.0 | 1.7 | 36.5 | 191.8 | 13.8 | 3.5 | −51.8 | 0.19 | 5.25 | 2.7 | 54.8 |
| 4 | 304 | 810 | 204 | 998 | 1.5 | 2.7 | 24.5 | 200.5 | 18.4 | 3.2 | −58.7 | 0.12 | 8.18 | 1.3 | 62.7 |
| 5 | 407 | 975 | 207 | 1130 | 2.0 | 2.4 | 32.8 | 241.3 | 18.6 | 3.6 | −69.5 | 0.14 | 7.35 | 1.8 | 67.0 |
| 6 | 532 | 1058 | 144 | 1440 | 3.7 | 2.0 | 42.9 | 261.9 | 13.0 | 4.6 | −73.0 | 0.16 | 6.10 | 3.3 | 56.9 |
| 7 | 426 | 544 | 126 | 869 | 3.4 | 1.3 | 34.4 | 134.7 | 11.4 | 2.8 | −33.4 | 0.26 | 3.92 | 3.0 | 48.1 |
| 8 | 229 | 652 | 129 | 950 | 1.8 | 2.8 | 18.5 | 161.4 | 11.6 | 3.0 | −47.6 | 0.11 | 8.74 | 1.6 | 53.8 |
| 9 | 230 | 878 | 130 | 1432 | 1.8 | 3.8 | 18.5 | 217.3 | 11.7 | 4.6 | −66.3 | 0.09 | 11.72 | 1.6 | 47.2 |
| 10 | 310 | 945 | 210 | 1490 | 1.5 | 3.0 | 25.0 | 233.9 | 18.9 | 4.8 | −69.6 | 0.11 | 9.36 | 1.3 | 48.7 |
| 11 | 455 | 1010 | 255 | 1519 | 1.8 | 2.2 | 36.7 | 250.0 | 23.0 | 4.9 | −71.1 | 0.15 | 6.81 | 1.6 | 51.0 |
| 12 | 275 | 190 | 75 | 708 | 3.7 | 0.7 | 22.2 | 47.0 | 6.8 | 2.3 | −8.3 | 0.47 | 2.12 | 3.3 | 20.4 |
| 13 | 399 | 454 | 99 | 885 | 4.0 | 1.1 | 32.2 | 112.4 | 8.9 | 2.8 | −26.7 | 0.29 | 3.49 | 3.6 | 40.1 |
| 14 | 240 | 672 | 140 | 945 | 1.7 | 2.8 | 19.4 | 166.3 | 12.6 | 3.0 | −49.0 | 0.12 | 8.59 | 1.5 | 55.4 |
| 15 | 119 | 107 | 19 | 135 | 6.3 | 0.9 | 9.6 | 26.5 | 1.7 | 0.4 | −5.6 | 0.36 | 2.76 | 5.6 | 66.3 |
| 16 | 225 | 150 | 25 | 142 | 9.0 | 0.7 | 18.1 | 37.1 | 2.3 | 0.5 | −6.3 | 0.49 | 2.05 | 8.1 | 74.2 |
| 17 | 228 | 190 | 28 | 147 | 8.1 | 0.8 | 18.4 | 47.0 | 2.5 | 0.5 | −9.5 | 0.39 | 2.56 | 7.3 | 94 |
| 18 | 333 | 178 | 33 | 170 | 10.1 | 0.5 | 26.9 | 44.1 | 3.0 | 0.5 | −5.7 | 0.61 | 1.64 | 9.0 | 88.2 |
| 19 | 432 | 185 | 32 | 168 | 13.5 | 0.4 | 34.8 | 45.8 | 2.9 | 0.5 | −3.7 | 0.76 | 1.31 | 12.1 | 91.6 |
| 20 | 337 | 204 | 37 | 180 | 9.1 | 0.6 | 27.2 | 50.5 | 3.3 | 0.6 | −7.8 | 0.54 | 1.86 | 8.2 | 84.2 |
| 21 | 219 | 47 | 19 | 212 | 11.5 | 0.2 | 17.7 | 11.6 | 1.7 | 0.7 | 2.0 | 1.52 | 0.66 | 10.3 | 16.6 |
| 22 | 222 | 94 | 22 | 245 | 10.1 | 0.4 | 17.9 | 23.3 | 2.0 | 0.8 | −1.8 | 0.77 | 1.30 | 9.0 | 29.1 |
| 23 | 233 | 88 | 33 | 298 | 7.1 | 0.4 | 18.8 | 21.8 | 3.0 | 1.0 | −1.0 | 0.86 | 1.16 | 6.3 | 21.8 |
| 24 | 239 | 125 | 39 | 306 | 6.1 | 0.5 | 19.3 | 30.9 | 3.5 | 1.0 | −3.9 | 0.62 | 1.61 | 5.5 | 30.9 |
| Permissible limit [6] | 50 | 50 | 50 | 500 | – | – | – | – | – | – | – | – | – | - | – |

For the amounts of natural radioisotopes $^{238}$U, $^{226}$Ra, $^{232}$Th, and $^{40}$K in the samples under investigation, the basic descriptive statistics are N, minimum, maximum, mean, standard deviation, skewness, kurtosis, and median, which are listed in Table 2. The table indicated that the distribution of the four radionuclides $^{238}$U, $^{226}$Ra, $^{232}$Th, and $^{40}$K is highly uniform, with the standard deviation values for each radionuclide being lower than their corresponding mean values [25].

**Table 2.** Statistical calculations of $^{238}$U, $^{232}$Th, $^{226}$Ra and $^{40}$K activity concentrations in the collected granite samples as well as comparison to another similar previous international research.

| Parameters | $^{238}$U (Bq.kg$^{-1}$) | $^{232}$Th (Bq.kg$^{-1}$) | $^{226}$Ra (Bq.kg$^{-1}$) | $^{40}$K (Bq.kg$^{-1}$) |
|---|---|---|---|---|
| N | 24 | 24 | 24 | 24 |
| Minimum | 119 | 47 | 19 | 135 |
| Maximum | 532 | 1058 | 255 | 1519 |
| Mean | 318.6 | 486.8 | 102.5 | 726 |
| Std. deviation | 105.1 | 352.9 | 73.4 | 496.5 |
| Skewness | 0.56 | 0.46 | 0.85 | 0.29 |
| Kurtosis | −0.90 | −1.74 | −1.12 | −1.58 |
| Median | 307 | 499 | 112.5 | 877 |
| **Countries Names, References** | $^{238}$U (Bq.kg$^{-1}$) | $^{232}$Th (Bq.kg$^{-1}$) | $^{226}$Ra (Bq.kg$^{-1}$) | $^{40}$K (Bq.kg$^{-1}$) |
| Egypt [26] | 137 | 82 | – | 1082 |
| Saudi Arabia [27] | 54.5 | 43.4 | – | 677.7 |
| Turkey [28] | 45.4 | 82.3 | – | 931.6 |
| Nigeria [29] | 74 | 100 | – | 1098 |
| China [30] | 355.9 | 317.9 | – | 1636.5 |
| Iran [31] | 38 | 47 | – | 917 |
| Italy [32] | 81.33 | 129 | – | 1065 |
| USA [33] | 31 | 61 | – | 1082 |
| Jordan [34] | 41.5 | 58.4 | – | 897 |

Peakedness is measured by kurtosis. Additionally, it is a result of internal distribution or sorting. It is described as mesokurtic, leptokurtic, or platykurtic, depending on how high the peak is. Kurtosis of zero is referred to as the normal curve or mesokurtic. Leptokurtic curves have more peaks than normal curves when the kurtosis value is positive; conversely, hypokurtic curves have fewer peaks than hypokurtic curves. The radionuclides $^{238}$U, $^{232}$Th, $^{226}$Ra, and $^{40}$K in the current investigation have leptokurtic kurtosis values that are negative (Table 2). This might be a result of the study area samples having an uneven distribution of natural radionuclides.

A frequency distribution's skewness is defined as its symmetry or lack of symmetry. Skewed distributions are described as being non-symmetrical distributions. Positively or negatively skewed distributions are also possible. The skewness of the activity concentrations of the radionuclides $^{238}$U, $^{232}$Th, and $^{40}$K was positive in the current study, demonstrating the asymmetry of their distributions. The displayed graph does not have a bell-shaped shape, and the positive values showed a positive skewness (Figure 4). The frequency distribution of the natural radioisotopes $^{238}$U, $^{232}$Th, $^{226}$Ra, and $^{40}$K is shown in Figure 3. The plotted graph has no bell-shaped form, and its positive values indicate a positive skewness. The frequency distributions of $^{238}$U, $^{232}$Th, $^{226}$Ra, and $^{40}$K were shown in Figure 4.

### 3.2. Environmental Hazard Impacts Results

The absorbed gamma dose rate (AGDR) compliance aids in the avoidance of deterministic effects and the control of potential stochastic impacts. All values were clearly above the permissible limit; 55 nGy.h$^{-1}$ [35,36]. The values of the AGDR ranged between 125.8 and 1375.7 nGy.h$^{-1}$, with an average value of 651.03 nGy.h$^{-1}$ (Table 3). It is worth mentioning that the collected granite samples exceed the permissible levels due to the presence of radioelement-bearing minerals such as zircon and monazite [35].

The annual effective dose rate values in the collected samples ranged between 0.2 and 1.7 mSv.y$^{-1}$, with an average of 0.8 mSv.y$^{-1}$ (Table 3). About 38% of the collected samples were lower than the permissible limit of the annual effective dose rate; 0.48 mSv.y$^{-1}$ and 62% of the collected samples exceeded this permissible limit [35]. Ra$_{eq}$ values for all studied samples were computed and are listed in Table 3. They fluctuated between 100.6 and 1795.5 Bq.kg$^{-1}$, with an average value of 844.46 Bq.kg$^{-1}$, which is higher than the recommended limit of 370 Bq.kg$^{-1}$ (Table 3) [36]. The variation of the Ra$_{eq}$ values is attributed to the mineralogical alteration processes affecting the studied rocks.

For the safe utilization of granitic rocks as building materials, H$_{ex}$ and H$_{in}$ ought not to be above unity [37]. H$_{ex}$ values ranged between 0.3 and 4.9, with an average value of 2.31 (Table 3), which falls higher than the permissible limit (unity); these results point to the fact that the collected rocks are harmful. Paradoxically, H$_{in}$ values ranged between 0.3 and 5.6, with an average value of 2.58 falling above unity (Table 3).

The alpha index, I$_{\alpha}$, is used for assessing the internal exposure level (the excess alpha radiation) as a result of inhaling radon gas arising from building materials. The alpha index takes into account that building materials having an activity concentration of radium should be less than 200 Bq.kg$^{-1}$; this is the recommended value [38]. I$_{\alpha}$ ranged between 0.1 and 1.3, with an average value of 0.52 (Table 3). The suggested permissible level of radium activity concentration for building materials is 100 Bq.kg$^{-1}$ (I$_{\alpha}$ = 0.5), while the upper level recommended for radium activity concentration is 200 Bq.kg$^{-1}$; I$_{\alpha}$ = 1 [39]. Accordingly, radiation risks are associated with these rocks if they are used in bulk quantities as construction material.

**Table 3.** Statistical parameters of environmental hazard indexes.

| Parameters | AGDR nG.h$^{-1}$ | AEDR | Ra$_{eq}$ | H$_{ex}$ | H$_{in}$ | I$_{\gamma}$ | ELCR | Iα |
|---|---|---|---|---|---|---|---|---|
| N | 24 | 24 | 24 | 24 | 24 | 24 | 24 | 24 |
| Minimum | 125.8 | 0.2 | 100.6 | 0.3 | 0.3 | 0.7 | 0.5 | 0.1 |
| Maximum | 1375.7 | 1.7 | 1795.5 | 4.9 | 5.6 | 12.8 | 5.6 | 1.3 |
| Mean | 651.03 | 0.80 | 844.46 | 2.31 | 2.58 | 6.04 | 2.64 | 0.52 |
| Median | 725.85 | 0.90 | 882.45 | 2.40 | 2.75 | 6.35 | 2.95 | 0.55 |
| Std. deviation | 452.63 | 0.56 | 604.08 | 1.65 | 1.85 | 4.30 | 1.84 | 0.37 |
| Skewness | 0.38 | 0.37 | 0.43 | 0.45 | 0.41 | 0.43 | 0.40 | 0.92 |
| Kurtosis | −1.74 | −1.74 | −1.77 | −1.76 | −1.76 | −1.77 | −1.71 | −0.98 |
| Permissible limit [6] | 55 | 0.48 | 370 | 1 | 1 | 2–6 | $0.29 \times 10^{-3}$ | 1 |

There should be no restrictions on the radioactivity of building materials. In terms of radiation protection, effective doses that are higher than the dose criterion of 0.48 mSv.y$^{-1}$ should be taken into consideration. As a result, it is suggested that controls be based on a dose range of 0.3 to 1 mSv.y$^{-1}$, which is the gamma dose contribution of building materials to the dose received outdoors [40]. The gamma index (I$_{\gamma}$) was calculated to see if our granitic samples meet those two standards. Materials for the decorative surface and other building uses, such as tiles, boards, and granite, meet the dose requirement for exemption, 0.3 mSv.y$^{-1}$, while the dose criterion of 1 mSv.y$^{-1}$ conforms to an I$_{\gamma}$ > 6 gamma representative level index. However, because these values correspond to annual effective dose rates greater than 1 mSv.y$^{-1}$ [40], samples with I > 6 cannot be used in construction. The studied granite samples' gamma index (I$_{\gamma}$) values were calculated. The average value was 6.04, with a range of 0.7 to 12.8 (Table 3). According to Table 3, it is clear that all excess lifetime cancer risk (ELCR) values were higher than the permissible limit of $0.29 \times 10^{-3}$. These values ranged from 0.5 to 5.6. The Pearson correlation method and hierarchical cluster analysis were used to effectively demonstrate the relationship between all of the radiological variables. Figure 5 depicts the relationship between the relevant environmental radiological parameters and the radionuclide concentrations in the dendrogram produced by the HCA. HCA is a data classification system that uses multivariate algorithms to determine real data groups. Objects are grouped in such a way that they all belong to

the same category. The results with the highest degree of nearness are categorized first in hierarchical clustering, followed by the next most similar data. The process continues until all of the information has been classified. The degrees of similarity at which the data mix is used to create a dendrogram. A similarity of 100% indicates that the clusters are divided by comparable sample measures by zero distance, whereas a similarity of 0% indicates the opposite. The clustering regions are as dissimilar as the least similar region. Three clusters were plotted in the dendrogram of the examined results for the Ha'il area. Cluster I consists of $^{238}$U, while Cluster II consists of $^{232}$Th, $^{226}$Ra, $^{40}$K, and one radiological hazard, the alpha index ($I_\alpha$). At the same time, Cluster III consists of all other radiological hazards. Thus, it can be concluded that the radioactivity and radiation exposure of Saudi Arabia granitic rocks used as building materials was linked mainly to the uranium, radium, thorium, and potassium activity concentrations.

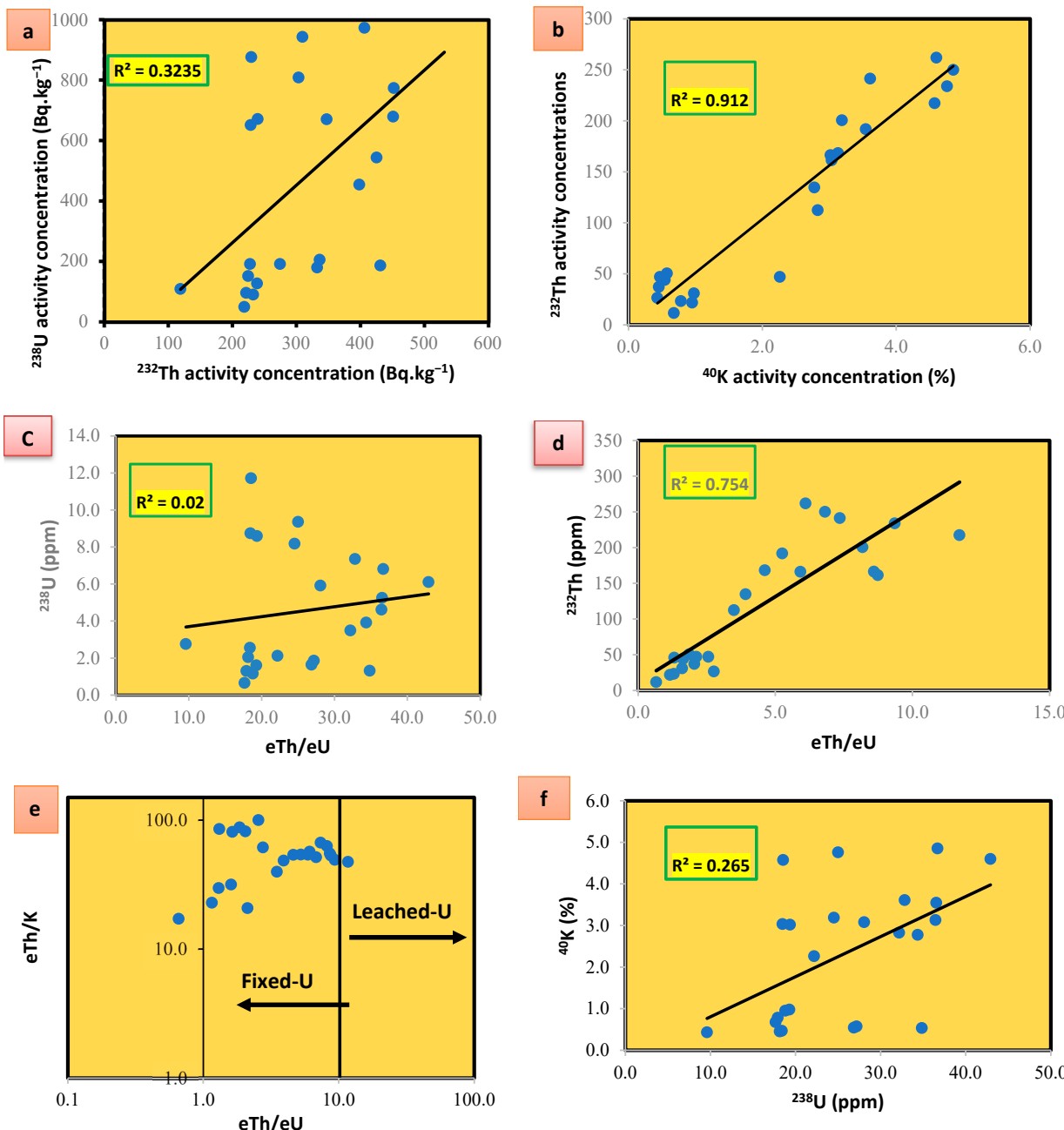

**Figure 2.** The radioactive correlations, (**a**) $^{238}$U vs. $^{232}$Th, (**b**) $^{232}$Th vs. $^{40}$K and (**c**) $^{238}$U vs. eTh/eU (**d**) $^{232}$Th vs. eTh/eU (**e**) eTh/K vs. eTh/eU (**f**) $^{40}$K vs. $^{238}$U.

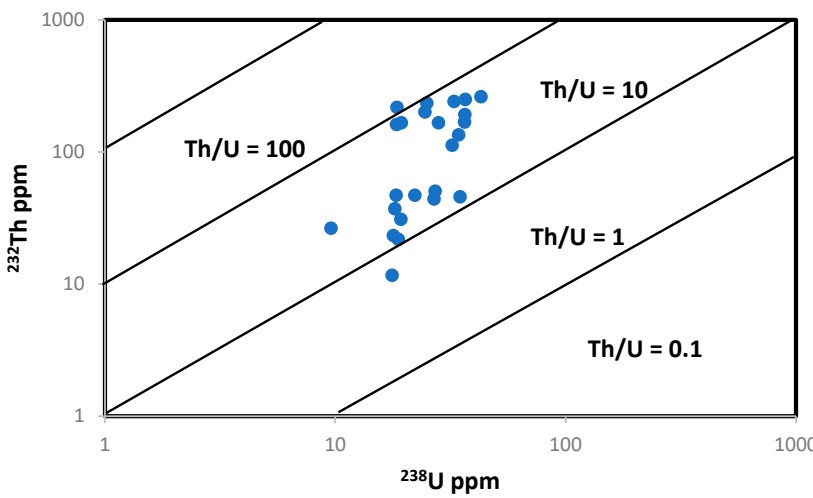

**Figure 3.** Th/U ratio variation diagram of the studied granitic samples.

**Figure 4.** Frequency distribution of [238]U, [232]Th, [226]Ra and [40]K.

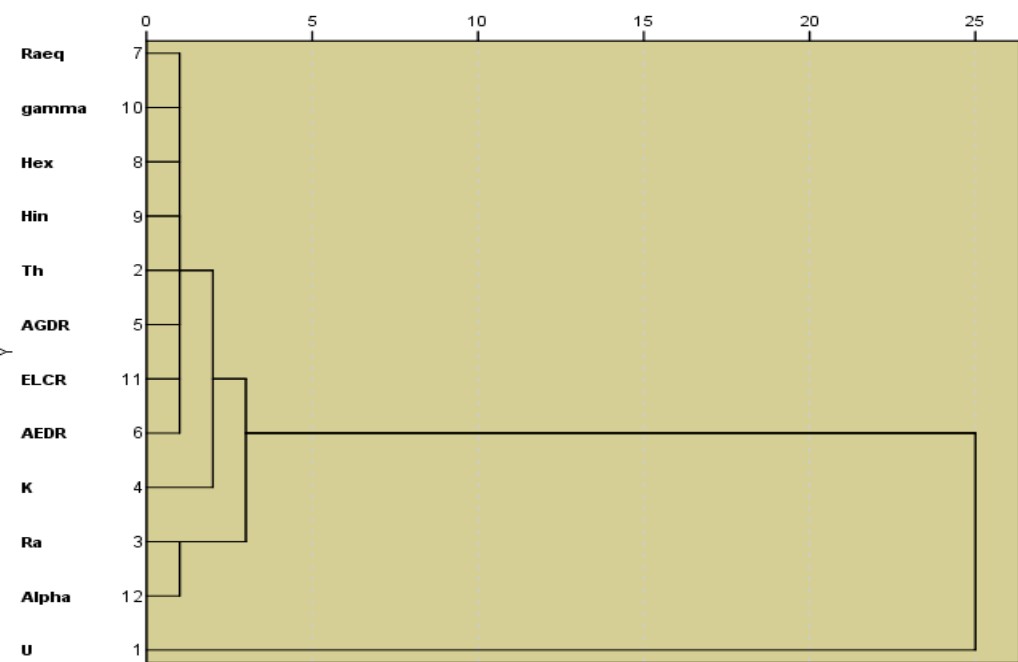

**Figure 5.** The dendrogram for the relationship among the studied radiological variables of the granites.

## 4. Conclusions

Twenty-four granitic rock samples were collected from the Hai'l area in Saudi Arabia. These granitic rocks were used as building materials. The activity concentrations of $^{238}$U, $^{232}$Th, $^{226}$Ra, and $^{40}$K were measured in the collected granite rock samples by using gamma spectrometry. The results of the activity concentrations were tabulated and compared with international standards. It was found that the average concentration values of the radioisotopes $^{238}$U, $^{226}$Ra, $^{232}$Th, and $^{40}$K were higher than the permissible limits according to UNESCEAR, 2000 [6]. Some environmental radiological hazard parameters were also calculated. Absorbed gamma dose rate values ranged between 125.8 and 1375.7 nGy.h$^{-1}$ with an average value of 651.03 nGy.h$^{-1}$. It is worth mentioning that the collected granite samples exceed the permissible limits due to the presence of radioelement-bearing minerals such as zircon and monazite. The Annual effective dose rate values in the collected samples ranged between 0.2 and 1.7 mSv.y$^{-1}$ with an average of 0.8 mSv.y$^{-1}$. About 38% of the collected samples were lower than the permissible limit of the annual effective dose rate of 0.48 mSv.y$^{-1}$, and 62% of the collected samples exceeded this permissible limit. Ra$_{eq}$ values for all studied samples fluctuated between 100.6 and 1795.5 Bq.kg$^{-1}$ with an average value of 844.46 Bq.kg$^{-1}$ which is higher than the recommended limit of 370 Bq.kg$^{-1}$. H$_{ex}$ values ranged between 0.3 and 4.9 with an average value of 2.31, which falls higher than the permissible limit (unity). Paradoxically, the H$_{in}$ values ranged between 0.3 and 5.6, with an average value of 2.58 falling above unity. I$_{\alpha}$ ranged between 0.1 and 1.3 with an average value of 0.52. Accordingly, radiation risks are associated with these rocks if they are used in bulk quantities as construction material. The gamma index; I$_{\gamma}$ values of the studied granite samples were calculated. The values of the gamma index ranged between 0.7 and 12.8 with an average of 6.04. It is clear that all values of ELCR were higher than the permissible limit of $0.29 \times 10^{-3}$ as these values ranged between 0.5 and 5.6.

**Author Contributions:** Conceptualization, O.F. and M.R.K.; methodology, M.R.K.; software, O.F.; validation, O.F., M.R.K.; formal analysis, O.F.; investigation, M.R.K., O.F.; resources, O.F., M.R.K.; data curation, O.F. and M.R.K.; writing-original draft preparation, O.F. and M.R.K.; writing-review and editing, O.F. and M.R.K.; visualization, M.R.K.; supervision, O.F. and M.R.K.; project administration, O.F. and M.R.K.; funding acquisition, O.F. All authors have read and agreed to the published version of the manuscript.

**Funding:** The authors extend their appreciation to the Deanship of Scientific Re-search (DSR), King Abdulaziz University, Jeddah under grant number IFPRC-194-135-2020.

**Data Availability Statement:** Not applicable.

**Acknowledgments:** This research was funded by the Deanship of Scientific Research (DSR), King Abdulaziz University, Jeddah under grant number; IFPRC-194-135-2020. The authors acknowledge the technical and financial support provided by the Ministry of Ed-ucation and King Abdulaziz University, Jeddah, Saudi Arabia, in the preparation of this paper.

**Conflicts of Interest:** The authors declare no conflict of interest.

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
