# Peer review of "Evaluation of Environmental Radioactivity and Hazard Impacts Saudi Arabia Granitic Rocks Used as Building Materials"

_minerals, doi:10.3390/min13020165_

Round 1

Reviewer 1 Report

The collected samples should test according to ASTM or EN standards, which will include: (1) Determination of water absorption at atmospheric pressure, (2)Determination of flexural strength under concentrated Load, (3) Determination of real density and apparent density, (4) Determination of Compressive Strength, (5) Determination of Abrasion Resistance, (6) Determination of flexural strength under constant moment, (7) Determination of the Breaking Load at Dowel hole, EN 13364, (8) Determination of Rupture Energy, (9) Determination of resistance to salt crystallization, EN 12370, (10) Determination of the slip resistance by means of the pendulum tester, EN 14231, (11) Determination of resistance to ageing by thermal shock, EN 14066

Author Response

The reviewer 1 asked to apply some tests to our samples and we don't have the facilities and the experience to apply these tests. He also mentioned that that it is not mandatory. We measured and study the radioactivity of these materials that used as building materials. 

Reviewer 2 Report

Thank you for the opportunity to review the manuscript (Evaluation of environmental radioactivity and hazard impacts of Saudi Arabian granitic rocks used as building materials) by Othman Fallatah and Mahmoud R. Khattab, submitted for possible publication in minerals.

The authors studied the distribution of natural radioactivity in commonly granitic rocks that are used as a building material along with the determination of environmental hazard impacts in Hai’l area (Saudi Arabia). The manuscript is interesting and represents a contribution to the radiological effect of granitic rocks.

The main comments are:

1.      There are some typo mistakes in the whole manuscript (see the attached Pdf file)

2.      Introduction is poor and lacks to present the topic of natural radioactivity from different sources as well as missing some relevant literature. I encourage the authors to rewrite and improve this section. I suggest reading some publications and recommend updating your references.

3.      The results in this manuscript should be compared with the others reported in the literature form various environmental and geological samples.

4.      I recommend rewriting and revising the conclusion section. In the list of references, please check and revise all the written references.

5. Further minor comments in the attached Pdf file

6.  Finally, the whole manuscript should be revised carefully to reach an adequate scientific level. The authors must have to make a big effort to improve the quality of their manuscript.

Author Response

Dear Sir

Thanks for your comments. I have corrected the paper according to your comments.

Best regards

Othman Fallatah

Reviewer 3 Report

Overall, the paper describes  very interesting and important aspect – natural radioactivity of rocks used for building materials production. Please, find below my comments:

Abstract- information about obtained results should be added in this section, in present form abstract is not informative

Lines 33-35- references should be added

Line 182 0.032 is higher than 0.3? Moreover, if the coefficient is 0.032 there is no correlation

Section 3.1 For what did you calculate correlation between e.g. thorium and potassium and how can you explain obtained value? For obtained coefficients should be add some comments.

Moreover, if you measured only rocks for what did you use Th/U ratio? Thorium and uranium have completely different properties and represents different decay chains, so how can you discuss its ratio? Fingerprint of what? Generally, the ratio (I mean Th/U) should be enhanced by other ratio (e.g.) if you measured radiatoactovity of e.g. water and sediments samples- using ratio in this case you have information about migration, washout etc. In presented by you form Th/U radio has no sense in my opinion. Additionaly it worth to mention that it’s really hard to washout Th. Moreover if you would like to discuss about U enrichment why didn’t you measure 234U??? The ratio between 234 and 238U is useful to describe and characterize uranium enrichment. And if the aim of your paper was measured natural radioactivity in rocks and hazard index and radiological safety  the more illogical is to use Th/U ratio.

Line 242- reference have to be add for the value of 0.48mSv/y

Lines 248-252- this part should be rewritten with additional comments

In line 242 permission limit is 0.48 mSv/y and in line 264 1 mSv/y. The value of 1 mSv/y is recommendation of IAEA and the value of 0.48 mSv/y is for what?

Line 276 where can I find Pearson coefficients?

Figure 5 there is no comments about this dendrogram in the paper. Moreover, in this case, PCA (principal component analysis) is better.

Author Response

Dear Sir

Thanks for your comments. I have corrected the papers according to your comments.

Best Regards

Othman Fallatah

Round 2

Reviewer 2 Report

The authors have provided responses to all of my queries and acted accordingly as per my suggestions and recommendations. 

The authors are requested to improve the resolution of figure 4.  Authors need to work on the quality of figures during proofreading. Also in Line 425 please correct (ref. 35)  from Mabrouk, S., .... to  Sami, M., ......

In general, the revised manuscript is an improved version in all aspects and I recommend it for publication. 

Reviewer 3 Report

Comprehensive study and an interesting topic the paper. The Authors revised the manuscript, and I believe it is suitable for publication in its present form.